# New Horizons with Growth Differentiation Factor 15 in Oncology: From Cancer Cachexia and Tumour Immunity to Novel Therapeutic Strategies

**DOI:** 10.3390/curroncol32110604

**Published:** 2025-10-29

**Authors:** Keiji Sugiyama, Naureen Starling, Ian Chau

**Affiliations:** 1Gastrointestinal Unit, Department of Medicine, Royal Marsden Hospital, London SW3 6JJ, UK; keiji.sugi@gmail.com (K.S.);; 2Department of Medical Oncology, NHO Nagoya Medical Center, Nagoya 460-0001, Aichi, Japan

**Keywords:** autonomic nervous system, cancer cachexia, growth differentiation factor 15, immune checkpoint inhibitors

## Abstract

Patients with cancer often experience weight loss, loss of muscle, poor appetite, and nausea, which reduce quality of life and make it harder to continue treatment. A protein called growth differentiation factor 15 is produced at high levels in many cancers. It acts on the brain to suppress appetite and trigger nausea, and it can also weaken the immune system, allowing tumours to escape control. This review explains the role of growth differentiation factor 15 in cancer, why it matters for patients, and how it may become a new target for treatment. We describe early clinical trials of medicines that block this pathway. These medicines have shown preliminary signs of benefit, including weight gain, improved appetite, reduced nausea, and potentially better responses to immunotherapy. Targeting growth differentiation factor 15 may help make cancer cachexia (a syndrome of ongoing weight and muscle loss in patients with cancer) go from an untreatable problem to a manageable condition, while improving both quality of life and treatment outcomes.

## 1. Introduction

Cancer cachexia is a multifactorial syndrome characterised by involuntary weight loss, muscle wasting, and metabolic dysfunction [1,2]. It affects most patients with advanced cancer, diminishing their quality of life (QoL) and response to anticancer therapies, ultimately leading to poor outcomes [2]. Despite extensive research, an established pharmacological therapy remains elusive [1,3,4].

Growth differentiation factor 15 (GDF-15) is a critical mediator of cancer cachexia, making it an attractive therapeutic target [5]. GDF-15, a stress-induced cytokine produced by peripheral tissues and/or tumour cells, exerts its effects by binding to the glial cell line-derived neurotrophic factor family receptor alpha-like (GFRAL) receptor in the brainstem [6]. GDF-15–GFRAL pathway activation induces anorexia and behavioural changes and enhances sympathetic nervous system (SNS) activity through the hypothalamus. Upregulated sympathetic outflow promotes a hypercatabolic state, driving lipolysis and skeletal muscle atrophy.

Certain platinum-containing regimens elevate GDF-15 levels, which mediates chemotherapy-induced nausea and vomiting (CINV) through mechanisms independent of known pathways [7]. GDF-15 plays a negative immunomodulatory role by impairing T-cell infiltration and promoting T-cell exhaustion [8], enabling cancer cells to escape immunity and resist immune checkpoint inhibitors (ICIs). These dual effects of metabolic dysregulation and immune escape make GDF-15 a highly attractive therapeutic target for reversing cancer cachexia, enhancing immunity, and improving systemic immunotherapy efficacy. Notably, elevated serum GDF-15 levels are associated with poor prognosis and cancer cachexia, including anorexia as a characteristic symptom across tumour types [9,10,11,12,13,14,15,16,17,18,19], highlighting its role in cancer biology.

Historically, clinical trials for cancer cachexia have evolved from empirical nutritional and anabolic interventions to targeted metabolic and hormonal approaches, such as ghrelin receptor agonists and selective androgen receptor modulators [4]. Recently, multimodal strategies incorporating nutrition, exercise, and anti-inflammatory therapy, exemplified by the MENAC trial [20], have represented an important step forward; however, current therapeutic strategies have shown only limited clinical benefit, highlighting the ongoing unmet need for mechanism-based and effective pharmacological interventions.

To comprehend the pathological implications of GDF-15 in cancer cachexia and immune modulation, examining its fundamental biological characteristics is essential. Therefore, this review provides a comprehensive overview of the emerging role of GDF-15 in cancer pathophysiology, focusing on its involvement in cachexia, immune suppression, and potential as a therapeutic target, supported by promising clinical results. The following section provides an overview of its physiological roles, regulatory mechanisms, and pathological overexpression, followed by discussions on its role in cancer.

## 2. Overview of GDF-15

GDF-15, also known as macrophage inhibitory cytokine-1 [21], exerts pleiotropic effects on metabolism, immune modulation, and homeostasis. It is a peptide consisting of 112 amino acids and belongs to the transforming growth factor-β (TGF-β) superfamily [22,23]. Although GDF-15 is expressed physiologically in the liver, pancreas, kidney, and placenta, its physiological functions are minimal; it does not significantly influence the energy balance under normal conditions [8].

GDF-15 is associated with hyperemesis gravidarum [24]. During gestation, GDF-15 is secreted by placental trophoblasts and contributes to nausea and vomiting, potentially promoting the avoidance of harmful foods or odours to protect the foetus [25]. Beyond its physiological roles, GDF-15 has been linked to obesity, insulin resistance, and type 2 diabetes, indicating broader metabolic relevance [26].

It is highly upregulated in response to cellular stress, inflammation, and tissue injury induced by hypoxia or DNA damage. GDF-15 is overproduced in the placenta during pregnancy and in diseases, including cancer, cardiovascular diseases, and metabolic disorders [22].

GDF-15 binds to GFRAL, which forms a complex with the receptor tyrosine kinase rearranged during transfection (RET). GFRAL is uniquely expressed in the area postrema (AP), known as the chemoreceptor trigger zone, and nucleus tractus solitarius (NTS), brainstem regions associated with appetite regulation and nausea [27]. However, GFRAL is aberrantly expressed in tumour cells [28] and peripheral tissues [29], suggesting a broader distribution than initially reported. GDF-15–GFRAL binding triggers a cascade of physiological actions, leading to appetite suppression and sympathetic activation. This promotes lipolysis and proteolysis to supply energy to peripheral organs from hepatic and skeletal muscle stores. These mechanisms are integral to acute phase responses to injuries or infections, commonly known as the ‘fight-or-flight response’ [30]. Nevertheless, excessive GDF-15 expression in chronic diseases, including advanced cancer, heart failure, [31] and kidney failure [32], induces deleterious effects, ultimately resulting in refractory conditions.

A recent phase II study (NIPICOL trial) in Microsatellite Instability–High or Deficient Mismatch Repair metastatic colorectal cancer (CRC) patients treated with nivolumab plus ipilimumab demonstrated that high baseline GDF-15 levels were associated with inferior progression-free and overall survival. Intriguingly, greater early reduction in GDF-15 levels was correlated with reversal of sarcopenia, suggesting a dual role of GDF-15 in both cancer progression and cachexia modulation [33].

GDF-15 also exerts an immunosuppressive effect on T lymphocytes and dendritic cells (DCs), preventing T-cell adhesion and migration into the tumour microenvironment (TME) [34]. Elevated GDF-15 levels in the TME inhibit the immune response against tumours, facilitating immune escape and contributing to ICI resistance [35]. The pathophysiological role of GDF-15 is presented in Figure 1.

In healthy individuals, GDF-15 circulates at 200–1000 pg/mL [5] and increases under pathological conditions [36]. GDF-15 concentrations were analysed using the Roche Elecsys assay in serum samples from healthy adults (n = 739) [37] and patients with cancer, including non-small cell lung cancer (NSCLC), pancreatic ductal adenocarcinoma (PDAC), and CRC, revealing elevated GDF-15 concentrations in all patients compared with healthy participants [38]. The mean GDF-15 levels for PDAC [39], CRC [18], NSCLC^16^, and oesophagogastric cancer [40] are 2990, 1371, 1258, and 1371 pg/mL, respectively. GDF-15 levels are associated with distant metastasis [41], anorexia, and weight loss [36,42]. The 95th percentile threshold of GDF-15 levels in healthy individuals is 1500 pg/mL, serving as a practical cutoff for pathological elevation.

### 2.1. Mechanism of GDF-15-Mediated Cancer Cachexia via the Neural System

GDF-15 can bind to GFRAL in the AP and NTS [22,43]. Lacking a blood–brain barrier, the AP detects circulating GDF-15 and induces appetite suppression [44]. The NTS receives afferent signals originating from visceral nerves that convey peripheral metabolic information [45], triggering nausea-like symptoms, fatigue, and reduced appetite and energy intake. NTS activation signals the hypothalamic satiety centre [46], suppressing appetite and stimulating SNS activity [6,43,45,47]. GDF-15 activates the hypothalamic–pituitary–adrenal (HPA) axis, leading to cortisol secretion [48]. Peripheral SNS activation promotes lipolysis in the liver and white adipose tissue via the β-adrenergic pathway. The resulting hypercatabolic state results in skeletal muscle proteolysis, contributing to muscle wasting [49]. Moreover, a decrease in adipose tissue and muscle volume is mediated by noradrenaline released from the end of peripheral sympathetic neurones or adrenaline from adrenal glands and β-adrenergic receptors [50]. In rodents, SNS denervation or β-adrenergic receptor antagonists attenuate GDF-15-mediated catabolic conditions without directly inhibiting GDF-15, demonstrating the involvement of neural pathways in GDF-15-induced cachexia via the central and peripheral nervous systems [51]. In a mouse cancer cachexia model, treatment with an anti-GFRAL antibody improves metabolic conditions [52]. The pathophysiological pathway linking abnormal GDF-15 secretion to involuntary weight loss and muscle wasting in patients with cancer has been elucidated, with elevated GDF-15 levels correlating with weight loss [36,53,54]. This extensive research suggests that increased GDF-15 levels in cancer cachexia may not just result from the condition but could also play a causative role in its development.

Although inflammatory cytokines, including tumour necrosis factor-alpha (TNF-α), interleukin-1 (IL-1), and IL-6, also play a role in cancer cachexia [55], their mechanisms differ from those of GDF-15. TNF-α is a pro-inflammatory cytokine that induces systemic inflammation through nuclear factor kappa-light-chain-enhancer of activated B cells signalling. It can directly affect the hypothalamus and cause flu-like symptoms [56]. Conversely, GDF-15 activates the SNS via the central nervous system without inducing systemic inflammation. GDF-15 acts through the GFRAL–RET pathway [6,43,57], whereas cytokines utilise the Janus kinase-signal transducer and activator of transcription pathway [56]. Both leptin [58] and GDF-15 can activate the SNS, promote lipolysis, and suppress appetite. In the hypothalamus, GDF-15 is thought to interact with key appetite-regulating neuroendocrine pathways, including inhibition of neuropeptide Y/agouti-related peptide (NPY/AgRP) neurons and activation of pro-opiomelanocortin (POMC) neurons, leading to anorexigenic effects. It may also functionally oppose ghrelin-mediated orexigenic signalling, although these cross-regulatory mechanisms remain incompletely defined [59,60,61]. GDF-15 plays a key role in anorexia and muscle wasting in patients with cancer, where various cytokines and humoral factors also contribute to these effects, acting through non-neural pathways and directly impacting peripheral organs [62].

### 2.2. Role of GDF-15 in CINV

CINV is a distressing side effect of systemic anticancer therapy [63]. Its mechanism is complex and involves the type of chemotherapy, patient-specific risk factors, and neurochemical pathways, particularly those mediated by 5-hydroxytryptamine type 3 (5-HT3) and neurokinin-1. Current standard treatments include 5-HT3 receptor antagonists, neurokinin-1 receptor antagonists, and corticosteroids. Despite significant progress in CINV management, it remains a serious issue, highlighting the need for novel therapies.

CINV is closely associated with the NTS and AP. Given the relationship between GDF-15 and appetite regulation, the role of GDF-15 in CINV has also been explored. Clinically, patients with germ cell tumours treated with a cisplatin-containing regimen exhibit increased GDF-15 levels [64,65]. In vivo, while cisplatin administration induces GDF-15 production, cisplatin-induced CINV and weight loss are not observed in *Gfral*-knockout mice. This suggests that GDF-15 is a key mediator of CINV and that GDF-15/GFRAL inhibitors could represent a new class of antiemetics [43,66]. Borner et al. comprehensively screened approved antiemetics for GDF-15-induced anorexia and found none to be effective, indicating that GDF-15-induced nausea is mediated through an independent mechanism. Recently, GFRAL antagonist [67] and anti-GDF-15 monoclonal antibody [68] have been tested in CINV animal models, demonstrating improvements in cisplatin-induced CINV. Furthermore, a randomised clinical trial of ponsegromab—an anti-GDF-15 antibody—in combination with chemotherapy demonstrated its potential efficacy in preventing CINV. In this trial, the incidence of nausea and vomiting was notably lower in the ponsegromab group than in the placebo group (nausea: 2% vs. 16%; vomiting: 4% vs. 13%) [38]. Although details of the background antiemetic regimen were not disclosed, these preliminary data suggest that GDF-15 inhibition may reduce CINV incidence in patients receiving cytotoxic chemotherapy. While the evidence remains early, these findings provide a hypothesis-generating signal warranting further clinical investigation.

### 2.3. Effects of GDF-15 in Tumour Immunity and TME

GDF-15, produced by tumour cells, can act in a paracrine manner within the tumour microenvironment, influencing surrounding stromal and immune cells. It promotes immune escape [34,69] by preventing T-cell adhesion and migration into the TME [34]. GDF-15 disrupts the leukocyte function-associated antigen-1–intercellular adhesion molecules-1 axis, which is critical for T-cell infiltration. High GDF-15 expression in tumours significantly reduces CD8+ cytotoxic and CD4+ T cells within the TME [35]. Furthermore, GDF-15 promotes regulatory T cell (Treg) differentiation from naïve CD4+ cells [70] and inhibits DC maturation [71]. These findings were confirmed using tissue from patients with refractory cancer and a history of immunotherapy, demonstrating that elevated serum GDF-15 concentrations (>1500 pg/ mL^3^) are associated with decreased cytotoxic CD3+ granzyme B+ T-cell density and proliferating CD3+Ki67+ T-cell density in the tumour. Furthermore, GDF-15 induces and activates myeloid-derived suppressive cells via the TGF/SMAD pathway and HPA axis [72]. This indicates that increased GDF-15 levels in the TME suppress the abundance, proliferation, and functionality of T-cell subsets in vivo [73].

Overall, GDF-15 plays a critical role in suppressing T-cell function and contributes to an ‘immune-cold’ tumour phenotype. This immunosuppressive function likely plays a protective role in certain physiological contexts [24,52]. Clinically, autoimmune disease activity diminishes during pregnancy, although the underlying mechanisms remain unclear [74]. The negative immunomodulatory effects of GDF-15, along with its elevated production by the placenta during pregnancy, may help explain this phenomenon. Conversely, elevated GDF-15 levels in the TME inhibit the host immune response against tumours, facilitating immune escape and contributing to resistance to immunotherapies, including ICIs [35]. ICIs are central to modern treatments of various cancer types. However, response rates to ICI monotherapy are typically modest or moderate [75,76,77,78], except in certain cancer types [79,80,81]. Therefore, combination therapies with chemotherapy, molecularly targeted therapies, or other immunotherapies have been widely clinically adopted or explored in clinical trials [82,83].

Preclinical and early clinical studies suggest that anti-GDF-15 therapy could represent a potential adjunct strategy to enhance immunotherapy efficacy [8,35]. Although it lacks intrinsic antitumour activity, GDF-15 inhibition may reduce Treg populations, restore cytotoxic T-cell function, and prevent their infiltration into the TME. These effects remain to be confirmed in larger trials but conceptually support the rationale for combination approaches. GDF-15 inhibition may promote DC maturation, further enhancing antitumour immunity.

Given the early stage of evidence, the clinical impact remains to be determined, and ongoing trials will clarify its translational relevance. Notably, cancer cachexia highlights the importance of preserving immune and nutritional fitness for therapeutic success [84,85,86,87,88]. Given its demonstrated anticachectic effects, anti-GDF-15 therapy could substantially address these challenges and improve clinical outcomes in patients with cancer cachexia [35].

### 2.4. Therapeutic GDF-15 Implementation

Based on the preclinical findings of the role of GDF-15 in cancer cachexia and tumour immunity, early-phase clinical trials have explored GDF-15 inhibition to manage cancer cachexia and enhance immunotherapy efficacy, demonstrating promising results (Figure 2).

Ponsegromab, a monoclonal antibody that neutralises GDF-15, is an actively studied agent for cancer cachexia. Two clinical trials (NCT03974776 and NCT03599063) have assessed its safety profiles in healthy volunteers. Ponsegromab was evaluated in two phase 1, single-dose studies [89]. Single subcutaneous doses of ponsegromab (0.1–300 mg) were well tolerated. Its toxicity profile was mild, with only four treatment-related adverse events (AEs) reported in three participants, which were limited to mild injection site reactions. A single ponsegromab administration at ≥1 mg effectively reduced serum unbound GDF-15 concentrations to below the lower detection limit of the assay within hours of administration.

A subsequent phase 1b study evaluated the safety and preliminary efficacy of ponsegromab in patients with advanced solid tumours who met Fearon’s criteria for cancer cachexia [2] and exhibited elevated serum GDF-15 concentrations (≥1500 pg/mL) [90]. Ponsegromab was subcutaneously administered (200 mg every 3 weeks for a maximum of 5 doses) over 12 weeks in 10 patients (4 with CRC, 3 with pancreatic cancer, and 3 with lung cancer). This treatment reduced serum GDF-15 levels below the detection limit. Patients gained an average weight of 4.63 kg compared with baseline and exhibited improvements in physical activity, skeletal muscle mass, and QoL. Notably, no ponsegromab-related AEs were reported. Ponsegromab was concurrently administered with first-line systemic anticancer therapies during the first or second cycle, following standard care. This suggests that ponsegromab demonstrates a favourable safety profile when combined with systemic anticancer therapies while effectively increasing body weight and mitigating anorexia under concurrent chemotherapy.

In a recent phase 2, randomised, double-blind trial, 281 patients with advanced cancer (NSCLC, PDAC, and CRC) were screened for cancer cachexia (defined by Fearon’s criteria [2]) and elevated serum GDF-15 levels (NCT05546476) [38]. A total of 187 patients were randomised, and among the 94 screen failures, 17 (6%) patients had GDF-15 levels below 1500 pg/mL. Considering the impact of chemotherapy on GDF-15 production and the effect of ponsegromab, with platinum-based chemotherapy, it was used as a stratification factor. Patients were randomly allocated to receive ponsegromab (100, 200, or 400 mg) or placebo subcutaneously every 4 weeks for 12 weeks, with 90% of patients also receiving systemic anticancer therapy, including 36% on platinum-based regimens. This trial demonstrated significant, dose-dependent weight gain at 12 weeks compared with the placebo group, with corresponding decreases in serum GDF-15 levels. Improvements were also observed in appetite and cachexia symptoms, total physical activity time, and muscle volume. AEs included anaemia, hypokalaemia, and diarrhoea and occurred at a similar frequency to that in the placebo group, suggesting that ponsegromab is a highly tolerable therapy for advanced cancer. A confirmatory phase 2b/3 study is anticipated to further clarify the impact of ponsegromab in patients with advanced solid tumours (NCT06989437). While these early findings are encouraging, they should be interpreted with caution, as the current evidence is limited to short-term, early-phase trials. Key questions to address include whether its anti-cachexia efficacy translates into improvements in the QoL and whether it confers a survival benefit.

AZD8853 is a monoclonal antibody that inhibits GDF-15. A phase 1/2 study investigated its dose-limiting toxicity (DLT), efficacy, and immune response in peripheral blood (NCT05397171). Among the 16 patients with pretreated NSCLC, CRC, or urothelial carcinoma (UC), 81.3% experienced AEs. The incidence of severe AEs was 37.5%; however, no treatment-related AEs or DLTs were observed [91,92]. Regarding efficacy, the disease control rate was 31.3%; however, no objective responses were observed, and the progressive disease rate was 68.8%. Pharmacokinetically, while AZD8853 has a half-life of 5–10 days, GDF-15 inhibition was not sustained. No clear evidence of a decrease in circulating tumour DNA or changes in peripheral immune cells, including CD8+ and CD4+ T cells, DCs, and myeloid-derived suppressor cells, was observed. Although AZD8853 exhibited a favourable safety profile, the study was terminated early owing to the lack of efficacy signals. Pharmacokinetic analyses and exploratory biomarkers suggested that transient suppression of circulating GDF-15, rapid target turnover, and lack of biomarker-based enrichment may have contributed to the absence of efficacy. The heavily pretreated population with immune-exhausted tumour microenvironments could also have limited the immunomodulatory potential of GDF-15 blockade. These findings highlight the pharmacological and biological challenges of translating GDF-15 inhibition into clinical benefit and underscore the need for sustained target coverage and rational patient selection in future studies.

Visugromab, a monoclonal antibody targeting GDF-15, has been used in refractory patients to ICIs. The phase 1–2a GDFATHER-1/2a trial (NCT04725474) investigated whether GDF-15 inhibition by visugromab can overcome immunotherapy resistance. Part A (phase 1, n = 25) enrolled refractory patients with advanced solid tumours who had been heavily pretreated with systemic therapy (a median of 4.4 lines), including ICIs, with no available established therapy. This phase evaluated the safety and tolerability of visugromab and nivolumab in a dose-escalation study, demonstrating the safety of visugromab without DLT and recommending a dose of 10 mg/kg for phase 2. Part B (phase 2a study) evaluated the antitumour activities of the combination in patients with solid tumours previously treated with ICIs. Among the 25 patients, 5 achieved stable disease or partial response, with a mean duration of response (DoR) of 12.9 months. Sequential biopsies demonstrated increased CD8+ T and CD4+FOXP3− T-cell infiltration and granzyme B (GZMB) overexpression, indicating tumour immunity activation by visugromab. Pan-cancer immunotranscriptomic analyses revealed a negative correlation between *GDF-15* mRNA expression levels and T-cell transcriptomic signatures in various solid cancer subtypes. Among these, NSCLC and UC were further evaluated in phase 2a. In the NSCLC cohort, 27 patients (21 non-squamous and 6 squamous) were enrolled. The overall response rate (ORR) was 14.8%, including two complete responses, all observed in NSCLC, which is consistent with translational research. The mean DoR was 15.3 months. In the UC cohort (n = 27), the ORR was 18.5%, with a mean DoR of 16.4 months. Among the responders, half exhibited significant tumour shrinkage compared to their initial ICI treatment. Visugromab plus nivolumab treatment induced CD8+ T, CD4+FOXP3− T, CD3+Ki67+ T, and CD3+GZMB+ T-cell infiltration and expansion in the TME. This combination also altered cytokine profiles, indicating interferon-gamma pathway activation. Baseline serum GDF-15 levels (cutoff > 1500 pg/mL) were significantly associated with decreased activated T cell and Treg densities. Decreased GDF-15 levels in tumour specimens were correlated with T-cell infiltration, suggesting an immunosuppressive effect of GDF-15 on tumour immunity. These findings suggest that visugromab may help reverse GDF-15–mediated immune suppression and restore sensitivity to ICIs, though confirmatory evidence from larger, randomised trials is awaited. The full results of the GDFATHER-2 study, including findings in various solid tumours (n = 273), have not been reported yet. Further randomised trials combining ICIs and standard systemic therapy as first- and second-line treatments are underway.

NGM120 is a monoclonal antibody that acts as an antagonist by binding to GFRAL. NGM120 binds to the GDF-15 receptor in the brainstem, inhibiting GDF-15-mediated effects. The GDF-15-mediated neural pathway may influence cancer cachexia and negatively affect tumour immunity via SNS activation [93,94]. Disrupting this pathway may inhibit the associated vicious cycle, potentially exerting antitumour activity. A favourable safety profile for NGM120 was demonstrated in a phase 1 study (NCT03392116, unpublished data). A subsequent phase 1a/1b evaluated the safety and preliminary antitumour activity of NGM120 (NCT04068896) in patients with metastatic PDAC. Phase 1a (n = 20) assessed NGM120 as a monotherapy, whereas phase 1b (n = 8) investigated its use in combination with gemcitabine and nab-paclitaxel. According to the phase 1a report, most AEs were grades 1–2. NGM120 increased lean body mass and decreased serum β-hydroxybutyrate, a marker of lipolysis. Overall, 25% of the patients had stable disease based on the Response Evaluation Criteria in Solid Tumours 1.1 criteria, although no objective response was observed. Phase 1b confirmed the feasibility of combining NGM120 with chemotherapy. All six radiologically evaluable patients achieved disease control at 16 weeks, including 3 partial responses, with 5 patients maintaining disease control beyond 32 weeks. An increase in lean body mass and weight gain was observed in patients who received NGM120 combined with chemotherapy. Previous and ongoing prospective trials are summarised in Table 1.

## 3. Discussion

This review highlights the pathophysiological roles of GDF-15 in cancer, with early-phase clinical trials suggesting its involvement in cancer cachexia and ICI resistance. We underscored the importance of neural networks in cancer cachexia, which can be targeted. Although the peripheral and central nervous systems, including humoral factors and inflammatory cytokines [55], play crucial roles in cancer cachexia, no treatments targeting them have been established. 5-HT3 antagonists [96] link clinical pharmacology to neuroscience in oncology; however, their application is limited to managing side effects. GDF-15-targeted therapy represents a novel approach for addressing neural mechanisms in cancer-associated symptoms and therapeutic implementation. The SNS suppresses tumour immunity, with a chronic hypersympathetic state being associated with poor prognosis [94,97,98,99]. GDF-15 inhibitors could mitigate this by reducing the impact of the SNS on tumour immunity. GDF-15 can be inhibited either directly or through blocking GFRAL using monoclonal antibodies. GDF-15 acts through two pathways: a neural mechanism involving the GFRAL–brainstem–peripheral nervous system and direct effects on immune cells, including T cells and DCs, independent of GFRAL. Compared with GFRAL inhibitors, GDF-15-directed therapy is a promising option for immunomodulation with ICIs, as it blocks these two pathways. While GFRAL receptor antagonists alleviate cachexia, appetite loss, and nausea, aberrant GFRAL expression has been reported in pancreatic [28] and gastric cancer cells [19]. Further research is warranted to clarify aberrant GFRAL expression in tumour tissues and the immunomodulatory effects of the GFRAL antagonists via neural pathways and direct effects on tumour cells, compared to anti-GDF-15 treatment.

GDF-15 influences chemotherapy efficacy. For instance, oxaliplatin-induced immunogenic reactions in CRC are mediated by suppressed GDF-15 secretion from tumour cells, activating tumour immunity [70]. Conversely, GDF-15 overproduction by gastric cancer cells mediates GFRAL signalling in tumour cells, resulting in cisplatin resistance [19]. These findings support the need for combination therapies targeting GDF-15/GFRAL with immunochemotherapy. Furthermore, GDF-15-mediated chemotherapy resistance has been reported for docetaxel and oxaliplatin in prostate cancer [100] and CRC [101], respectively. However, some clinical questions remain unresolved. First, the GDF-15 antibody demonstrated significant anti-cachectic efficacy, unlike the other agents targeting TNF-α [102,103,104] or thalidomide [105]. GDF-15 primarily regulates appetite and cell metabolism, and its targeted treatment can directly mediate cancer cachexia. Conversely, inflammatory cytokines, such as TNF-α or IL-6, exert systemic effects, affecting metabolism, appetite regulation, systemic inflammation, and immune response to pathogens. Therefore, anorexic effects mediated by TNF-α may not be central to cachexia. Second, the standardisation of non-pharmacological interventions, including nutritional support and personalised exercise regimens, is critical in cancer cachexia clinical trials. A high-protein diet with tailored exercise appears essential for maintaining or improving muscle volume and, ultimately, functional status [3,106,107]. Although pharmacological therapies lead therapeutic innovation, a multimodal approach is necessary. The MENAC study, a phase 3 randomised trial, evaluated the efficacy of a multimodal intervention comprising exercise, nutrition, and nonsteroidal anti-inflammatory drugs with standard care compared to standard care alone in patients with advanced cancer receiving chemotherapy [20]. This trial demonstrated significant body weight stabilisation in the intervention arm, highlighting the central role of multimodal support in managing cancer cachexia. In parallel, efforts to harmonise outcome measures in cancer cachexia trials are underway, as exemplified by the Cancer Cachexia Endpoints Working Group, which aims to improve consistency and clinical relevance in endpoint selection. Consequently, cancer cachexia trials should integrate standardised multimodal care into control and experimental arms.

Cancer cachexia progresses in three stages: pre-cachexia, cachexia, and refractory cachexia [2]. The therapeutic window lies in the first two stages, observed in metastatic cancer and locoregional disease [108]. Cancer cachexia is linked to a poor postoperative prognosis [109,110,111,112,113,114] and reduced response to ICIs [84,85,86,87,88]. Therefore, GDF-15 or GFRAL inhibitors should be evaluated in patients with non-metastatic disease to improve their prognosis. As neoadjuvant or adjuvant immunotherapy becomes standard, the dual benefits of GDF-15 inhibition, including mitigating cancer cachexia and enhancing ICI immunocompetency, make this approach an attractive strategy alongside chemotherapy and/or immunotherapy. Notably, the visugromab-nivolumab combination is currently under investigation for patients with bladder cancer (NCT06059547). Expanded evaluations in operable patients, particularly those with gastrointestinal cancers, including oesophagogastric and pancreatic cancers, where cachexia is highly prevalent, would be valuable.

GDF-15-targeted therapies should be evaluated using a tumour-agnostic approach to treat cachexia and enhance immunotherapy efficacy. A phase 2 study of ponsegromab enrolled patients with elevated GDF-15 levels and selected tumour types. Given the high cachexia prevalence in upper gastrointestinal, hepatobiliary, and head and neck cancers, further evaluation in these cancers [108] or through a tumour-agnostic approach is warranted [115].

Translational data from the GDFATHER-1 and GDFATHER-2a trials revealed an inverse correlation between elevated *GDF-15* mRNA expression in tissue samples and immunosuppressive TMEs [73]. These studies supported the preliminary success of visugromab plus nivolumab, which exhibited promising antitumour activity against specific tumour types. Some tumour types demonstrated a positive correlation between *GDF-15* mRNA levels and immune signatures. As GDF-15-mediated cachexia may independently suppress tumour immunity or diminish immunotherapy efficacy, these findings warrant evaluation to determine whether GDF-15 inhibition can enhance ICI efficacy across or within selected tumour types.

In summary, GDF-15/GFRAL antagonism may offer multiple therapeutic benefits, including anti-cachexia effects, potential relief of CINV, and enhanced immunotherapy. These combined effects can improve the prognosis and QoL of patients with cancer, warranting investigation across different cancer stages, types, and treatments. Preclinical studies highlight the critical role of the GDF-15/GFRAL pathway in cancer cachexia, with its inhibition mitigating symptoms. These findings are clinically supported in patients with advanced cancer, positioning GDF-15 as a promising example of bench-to-bedside translation from biological discovery to early clinical application. In addition to their potential for treating cancer cachexia, multiple ongoing and planned clinical trials are evaluating GDF-15 and GFRAL inhibitors in combination with ICIs. Targeting GDF-15 may help redefine the management of cancer cachexia as a potentially treatable condition, while also enhancing immunotherapy efficacy and expanding therapeutic strategies.

## Figures and Tables

**Figure 1 curroncol-32-00604-f001:**
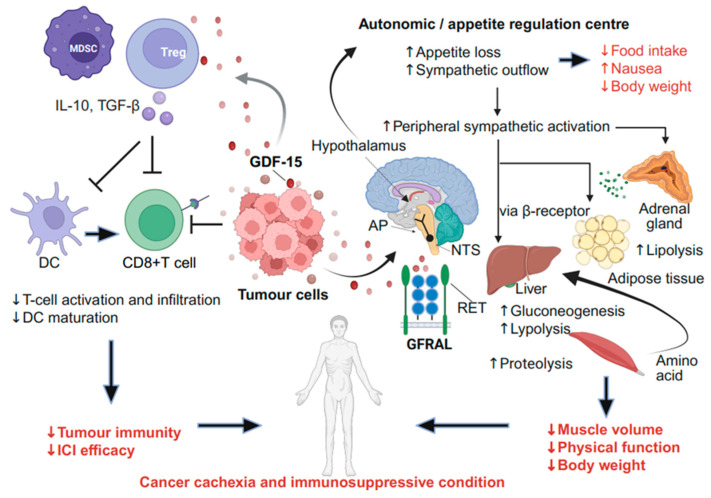
Mechanism of GDF-15-mediated cancer cachexia and suppression of tumour immunity. GDF-15 is produced by tumour cells and interferes with immune cells in the tumour microenvironment through paracrine signalling. GDF-15 inhibits T-cell infiltration and activation through direct or indirect mechanisms, leading to immune escape. It is transported via circulation to the AP (vomiting centre) and the NTS of the brainstem, where its receptor, GFRAL, is expressed. This process mediates cachexia related symptoms (nausea and anorexia) and activates the sympathetic nervous system, ultimately leading to the development of cancer cachexia. Furthermore, activated sympathetic flow interferes with tumour immunity by exerting an inhibitory effect (not illustrated here). This figure was created using BioRender (https://BioRender.com). AP, area postrema; DC, dendritic cell; GDF-15, growth differentiation factor 15; GFRAL, GDNF family receptor alpha-like; IL-10, interleukin-10; NTS, nucleus tractus solitarius; RET, rearranged during Transfection; TGF-β, transforming growth factor beta. Arrows indicate the direction of biological effects or signalling flow, whereas T-shaped lines represent inhibitory interactions.

**Figure 2 curroncol-32-00604-f002:**
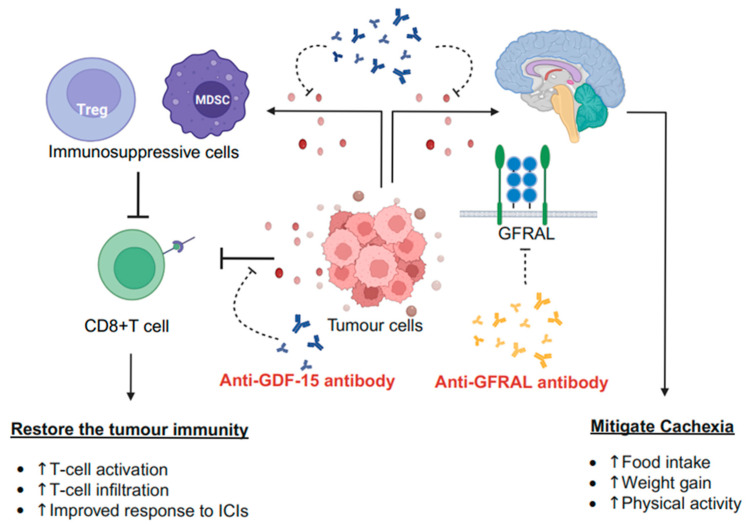
The mechanism underlying anti-GDF-15/GFRAL therapy in cancer. An anti-GDF-15 antibody binds to GDF-15 and inhibits its effects on the immune system and CNS, whereas an anti-GFRAL antibody blocks GFRAL, the specific receptor for GDF-15. The anti-GFRAL antibody can interfere with the GFRAL-mediated pathway. Preclinical data suggest that the aberrant expression of GFRAL on tumour cells and the peripheral nervous system modulates tumour immunity; this can be influenced by GFRAL antagonism (not illustrated here). Chemotherapy, such as cisplatin, induces GDF-15 overproduction and leads to nausea and vomiting via the CNS. Anti-GDF-15/GFRAL inhibitors are expected to alleviate chemotherapy-related symptoms. This figure was created using BioRender (https://BioRender.com). CNS, central nervous system; GDF-15, growth differentiation factor 15; GFRAL, glial cell line-derived neurotrophic factor family receptor alpha-like; ICIs, Immune Checkpoint Inhibitors; MDSC, Myeloid-Derived Suppressor Cell; Treg, Regulatory T cell. Arrows indicate the direction of biological effects or signalling flow, whereas T-shaped lines represent inhibitory interactions.

**Table 1 curroncol-32-00604-t001:** Summary of clinical trials targeting GDF-15/GFRAL.

Trial Identifier	Agent (Agent Type)	Target Molecule	Study Status	Study Design	Number of Patients	Target Population	Treatment	Inclusion Criteria by GDF-15 Levels	Primary Endpoint	**Results**
NCT04815551	AV-380 (mAB)	GDF-15	Completed	I, randomised, double-blind, placebo-controlled	56	Healthy volunteers	AV-380 vs. placebo	No	Safety, PK/PD analysis	Not reported
NCT05865535			Recruiting	I, nonrandomised	30	CRC, PDAC	AV-380 + standard chemotherapy	Yes (serum GDF-15 levels ≥ 1200 pg/mL)	Safety, PK/PD analysis	Not reported.
NCT05397171 [91]	AZD8853 (mAB)	GDF-15	Terminated	I/II, nonrandomised	16 (Part A)	NSCLC, CRC, UC	AZD8853	No	Safety	No DLT or safety concern was observed. No radiological tumour response. Only Part A was initiated. The entire Master Protocol was terminated.
NCT03392116	NGM120 (mAb)	GFRAL	Completed	I, randomised, double-blind, placebo-controlled	92	Healthy volunteers	NGM120 vs. placebo	No	Safety	Not reported.
NCT04068896 [95]			Completed	I/II, randomised, double-blind, placebo-controlled	75	PDAC, CRPC	Chemotherapy plus NGM120 or placebo	Yes (serum GDF-15 levels ≥ 1300 pg/mL [prostate cancer, part 3])	Safety	(Phase 1a/1b, PDAC with GEM + nab-PTX) No safety concern. Body weight gain was observed in monotherapy and combination with chemotherapy. DCR: 25% (5/20) with no objective response.
NCT03974776	Ponsegromab (mAb)	GDF-15	Completed	I, randomised, double-blind, placebo-controlled	8	Healthy volunteer (Japanese only)	Ponsegromab vs. placebo	No	Safety	Not reported
NCT04299048 [90]			Completed	IB, nonrandomised	10	NSCLC, PDAC, CRC	Ponsegromab	Yes (serum GDF-15 levels ≥ 1500 pg/mL)	Safety	A favourable safety profile was demonstrated. Body weight gain and improved physical activities, and appetite were observed. Inhibition of serum GDF-15 was confirmed at the lower detection limit of the assay.
NCT05546476 [38]			Active, not recruiting	II, randomised, double-blind, placebo (PROACC-1 study)	187	NSCLC, PDAC, CRC with elevated serum GDF-15	Ponsegromab vs. placebo	Yes (serum GDF-15 levels ≥ 1500 pg/mL)	Change from baseline in body weight at week 12	Significant body weight gain was observed. Improved appetite, cachexia symptoms, and physical activity were also observed (400 mg group).
NCT03599063 [89]			Completed	I, randomised, double-blind, placebo-controlled	63	Healthy volunteer	Ponsegromab vs. placebo	No	Safety	A favourable safety profile was demonstrated. Inhibition of serum GDF-15 was confirmed at the lower detection limit of the assay.
NCT04803305 [89]			Completed	I, randomised, double-blind, placebo-controlled	18	NSCLC, PDAC, CRC, prostate, breast	Ponsegromab vs. placebo	Yes (serum GDF-15 levels ≥ 1500 pg/mL)	Safety	A favourable safety profile was demonstrated. Inhibition of serum GDF-15 was confirmed at the lower detection limit of the assay.
NCT04725474 [73]	Visugromab (CTL002) (mAB)	GDF-15	Active, not recruiting	I/II, nonrandomised (GDFFATHER-2)	155	Advanced cancer with refractoriness to ICIs (NSCLC, UC, others)	Visugromab (CTL002) plus nivolumab	No	Safety, antitumour activity	No DLT was observed. ORR: NSCLC, 4/27 (14.8%), including 2 CR; UC, 5/27 (18.5%), including 1 CR
NCT06059547			Recruiting	II, randomised (GDFFATHER-NEO)	30	T2-T4aN0M0 MIBC (cisplatin-ineligible)	Visugromab (CTL-002) plus nivolumab vs. nivolumab	No	Antitumour activities (pCR rate, radiological tumour response rate)	Not reported

CRPC, castration-resistant prostate cancer; CRC, colorectal cancer; DCR, disease control rate; DLT, dose-limiting toxicity; GEM, gemcitabine; GDF-15, growth differentiation factor 15; GFRAL, glial cell line-derived neurotrophic factor family receptor alpha-like; mAb, monoclonal antibody; nab-PTX, nanoparticle albumin-bound paclitaxel; NSCLC, non-small cell lung cancer; PDAC, pancreatic ductal adenocarcinoma; UC, urothelial carcinoma.

## Data Availability

This article is a review and does not include original data.

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
