# Peer review of "New Horizons with Growth Differentiation Factor 15 in Oncology: From Cancer Cachexia and Tumour Immunity to Novel Therapeutic Strategies"

_curroncol, 2025, doi:10.3390/curroncol32110604_

Round 1
Reviewer 1 Report
Comments and Suggestions for Authors
Current Oncology Review
Full Title: New Horizons with Growth Differentiation Factor 15 in Oncology: From Cancer Cachexia and Tumour Immunity to Novel 3 Therapeutic Strategies
Summary:
This is a comprehensive review that summarizes the role of GDF15 in cancer cachexia, CINV, cancer immunity, and as a therapeutic target. The authors synthesize basic mechanistic data and emerging clinicals trials (ponse, NGM120, SAX885, and visu). The review highlights that the GDF15-/GFRAL pathway as a promising target, suggesting it might transform cachexia management while potentially enhancing immunotherapy effect.
The authors are to be acknowledged for being well-organized, including 2 mechanistic figures, and an up-to-date clinical trials table. The topic is certainly timely given ongoing later phase studies in this space and offers strong translational relevance. However, certainly clarifications and improvements would strengthen clarity, balance/hype, and clinical applications.
Suggested important revisions
Balance: The review, like most in this space, is heavily based on early phase clinical trial data but the limitations and known failures are under-emphasized. Ie. AZD8853 was terminated due to a lack of effect. A more balanced discussion on why such an agent (with similar MOA to other agents currently under investigation) would help contextualize the field and discussion.
Mechanistic clarification: The authors distinguish GDF15 from inflammatory cytokines but would benefit from highlighting how GDF-15 diverts and overlaps w these established cachexia pathways. The current version of the manuscript appears to overemphasize the singular role of GDF15.
Clinical trial review: The available (published data) notes improvement in weigh, anorexia, and QOL w ponsegromab but survival effects remain unclear clear. Strongly consider that despite weight loss being clearly prognostic for outcomes it is unknown if improving weight leads to improved survival (I certainly think and hope so) but should be noted explicitly that such data is not available. The visu+Nivo did show response rates, but these were modest at best and expectations for any agent w minimal single agent activity should be tempered in such a review. We have very little data if any that anti-GDF15 agent leads to response in the absence of chemo and/or immunotherapy and a vast hx of oncology studies have revealed that agents w poor single agent activity tend to not translate into “revolutionary” effects when combined w other modestly effective therapeutics.
CINV discussion: The mechanistic explanation and figures are strong. I think it would help to contextualize the GSDF-15 available data for CINV relative to other novel CINV agents such as NK1 antagonist. Your team provided the data (2 vs 16%) but was that with the use of standard guideline concordant CINV regimen etc or not?
Suggested Major revisions
Abstract and summary ad discussion: There are some lines that seem to overstate the field and therapeutic progress. Though I too remain hopeful, I think softer language would be more balanced and honest. (i.e., “transform cancer cachexia from untreatable to manageable” is bold). Similarly, consider switching “transformative strategy” to promising or emerging strategy. Line 431, “rapidly emerging as adjuncts to ICIs”, it has not emerged once yet beyond early phase trials. I certainly hope it will but consider “multiple ongoing and planned clinical trials will evaluate these agents with ICI therapeutics”…
Intro: Would clarify in line 53, that some chemotherapy rather than “Chemotherapy elevates GDG-15 levels,…”
Formatting/Spelling: Line 110 starts w “Graph 15”, likely typo and should be GDF-15. Line 121 “ca-chexia-related” seems like it should cachexia (without the hyphen). “Immune evasion” and immune escape are used, consider being consistent throughout. Author affiliation notes “Centreer”
Author Response
Commet1
Summary:
This is a comprehensive review that summarizes the role of GDF15 in cancer cachexia, CINV, cancer immunity, and as a therapeutic target. The authors synthesize basic mechanistic data and emerging clinicals trials (ponse, NGM120, SAX885, and visu). The review highlights that the GDF15-/GFRAL pathway as a promising target, suggesting it might transform cachexia management while potentially enhancing immunotherapy effect.
The authors are to be acknowledged for being well-organized, including 2 mechanistic figures, and an up-to-date clinical trials table. The topic is certainly timely given ongoing later phase studies in this space and offers strong translational relevance. However, certainly clarifications and improvements would strengthen clarity, balance/hype, and clinical applications.
Response: We sincerely thank the reviewer for the positive and constructive feedback.
We appreciate the recognition of the manuscript’s organization, clarity, and relevance, and have carefully revised the text to address each of the reviewer’s insightful comments.
Point-by-point responses are provided below.
Suggested important revisions
Comment 2:
Balance: The review, like most in this space, is heavily based on early phase clinical trial data but the limitations and known failures are under-emphasized. Ie. AZD8853 was terminated due to a lack of effect. A more balanced discussion on why such an agent (with similar MOA to other agents currently under investigation) would help contextualize the field and discussion.
We fully agree. The section on AZD8853 (Line 307) has been expanded to explicitly state that this program was terminated early due to lack of efficacy signals, despite acceptable safety. We also discuss the pharmacological and biological factors that may have contributed, including limited GDF-15 suppression and short drug half-life.
This revision provides a more balanced context regarding translational challenges and helps to temper expectations for ongoing programs with similar mechanisms.
Comment 3:
Mechanistic clarification: The authors distinguish GDF15 from inflammatory cytokines but would benefit from highlighting how GDF-15 diverts and overlaps w these established cachexia pathways. The current version of the manuscript appears to overemphasize the singular role of GDF15.
Response: We have added explicit discussion (Line 165) comparing GDF-15 signaling to inflammatory cytokines.
Specifically, the revised text now notes that GDF-15 induces cachexia primarily through central nervous system activation of the sympathetic nervous system, in contrast to TNF-α and IL-6, which mediate systemic inflammation through JAK/STAT and NF-κB pathways. This addition clarifies both the overlap and divergence of these mechanisms and places GDF-15 within the broader context of cachexia biology.
Comment 4: Clinical trial review: The available (published data) notes improvement in weigh, anorexia, and QOL w ponsegromab but survival effects remain unclear clear. Strongly consider that despite weight loss being clearly prognostic for outcomes it is unknown if improving weight leads to improved survival (I certainly think and hope so) but should be noted explicitly that such data is not available. The visu+Nivo did show response rates, but these were modest at best and expectations for any agent w minimal single agent activity should be tempered in such a review. We have very little data if any that anti-GDF15 agent leads to response in the absence of chemo and/or immunotherapy and a vast hx of oncology studies have revealed that agents w poor single agent activity tend to not translate into “revolutionary” effects when combined w other modestly effective therapeutics.
Response: We agree and have revised the relevant sections to explicitly acknowledge that current clinical data are limited to short-term, early-phase trials.
We state that while ponsegromab improved weight and symptoms, no data yet demonstrate survival benefit. Similarly, for visugromab, we describe the modest objective response rates and emphasize that confirmatory randomized studies are ongoing. These changes address the reviewer’s point and provide a realistic perspective on current efficacy evidence.
Comment 5:
CINV discussion: The mechanistic explanation and figures are strong. I think it would help to contextualize the GSDF-15 available data for CINV relative to other novel CINV agents such as NK1 antagonist. Your team provided the data (2 vs 16%) but was that with the use of standard guideline concordant CINV regimen etc or not?
Response: We have clarified this in Section 2.2 (Line 202). The revised text notes that while standard NK1-based antiemetic use was not detailed in the referenced study, the findings still suggest that GDF-15–mediated nausea and anorexia may act through mechanisms distinct from conventional pathways. This contextualizes the data relative to existing antiemetic strategies as suggested.
Suggested Major revisions
Comment 6: Abstract and summary ad discussion: There are some lines that seem to overstate the field and therapeutic progress. Though I too remain hopeful, I think softer language would be more balanced and honest. (i.e., “transform cancer cachexia from untreatable to manageable” is bold). Similarly, consider switching “transformative strategy” to promising or emerging strategy. Line 431, “rapidly emerging as adjuncts to ICIs”, it has not emerged once yet beyond early phase trials. I certainly hope it will but consider “multiple ongoing and planned clinical trials will evaluate these agents with ICI therapeutics”…
Response:
We have revised all such expressions throughout the Abstract and Discussion for a more balanced tone.
-
“Transform cancer cachexia from untreatable to manageable” → revised to “may help redefine the management of cancer cachexia as a potentially treatable condition.”
-
“Transformative strategy” → replaced with “promising strategy.”
-
The final paragraph now reads:
“Multiple ongoing and planned clinical trials are evaluating GDF-15 and GFRAL inhibitors in combination with immune checkpoint inhibitors (ICIs).”
These modifications align with the reviewer’s suggestion to reflect cautious optimism while maintaining scientific accuracy.
Comment 7:
Intro: Would clarify in line 53, that some chemotherapy rather than “Chemotherapy elevates GDG-15 levels,…”
Formatting/Spelling: Line 110 starts w “Graph 15”, likely typo and should be GDF-15. Line 121 “ca-chexia-related” seems like it should cachexia (without the hyphen). “Immune evasion” and immune escape are used, consider being consistent throughout. Author affiliation notes “Centreer”
Response: I edited the sentence as "Some chemotherapy regimens, such as platinum-based combinations, elevate GDF-15 levels.”
We also corrected all typographical issues and standardized the terminology to “immune evasion.”"
We thank the reviewer again for their constructive guidance, which significantly improved the balance, precision, and clinical relevance of our review.
Reviewer 2 Report
Comments and Suggestions for Authors
In this review article, Sugiyama et al. summarize the current evidence regarding the dual role of GDF-15 in cancer cachexia and tumour immunity. The manuscript is well-written and incorporates, in general, the current knowledge regarding the aforementioned associations.
Major comment.
A brief presentation of the evolution of cachexia clinical trials should be presented after the introduction.
Some minor comments are:
- Introduction – line 44: “…an established medical therapy…” The term “pharmacological therapy” is more accurate.
- Introduction – line 47: “…produced by peripheral tissues or tumour cells.” This should be replaced by “…peripheral tissues and/or tumour cells”
- Introduction – line 60: “…associated with poor prognosis and cancer cachexia or anorexia…” As anorexia is a symptom of the syndrome cachexia, this sentence needs to be rephrased.
- Overview of GDF15. A clearer presentation of the correlations of GDF15 with obesity, insulin resistance, and diabetes is needed.
- Overview of GDF15 – line 73: “Although GDF-15 is expressed …” Please provide a relevant ref.
- Overview of GDG 15: The third paragraph, “In healthy individuals…” which mostly discusses GDF15 levels and their associations in various cancers, should be transferred last (after the interpretation of figure 1)
- Overview of GDG 15 – line 110: “Graph 15. also exerts …” Is this a typo?
- Mechanism of GDF-15-mediated cancer cachexia via the neural system: In addition to leptin, the interrelation(s) of GDF-15 with other regulatory hormones (i.e. ghrelin) and neurotransmitters (i.e., neuropeptide Y, pro-opiomelanocortin) should also be discussed.
- Effects of GDF-15 in tumour immunity and TME – line 187: “can spread to adjacent tumour tissue…” Do the authors mean tumor microenvironment?
- Therapeutic GDF-15 implementation – line 273: “A confirmatory phase 3 study is anticipated…” The referred study (NCT06989437) is a phase IIb/III study.
- Table 1. For a more convenient data depiction, some columns (i.e., “agent” and “Type agent” and/or “study status” and “Results”) might be merged.
- Discussion – line 354: “The SNS suppresses tumour immunity suppressively, …” This sentence needs to be rephrased.
Author Response
Major comment
Comments:
A brief presentation of the evolution of cachexia clinical trials should be presented after the introduction.
Response:
We appreciate this valuable suggestion. A new paragraph has been added after the Introduction summarizing the historical evolution of clinical trials in cancer cachexia.
The revised section outlines the progression from early empirical nutritional and anabolic interventions to targeted metabolic and hormonal approaches (e.g., ghrelin receptor agonists and selective androgen receptor modulators), culminating in recent multimodal strategies such as the MENAC trial.
We have also noted that the MENAC trial demonstrated positive results—a remarkable achievement in this challenging field—yet the clinical benefit to patients remains minimal. Therefore, further development, particularly of mechanism-based pharmacological interventions, remains essential.
This addition provides historical and conceptual context for current GDF-15–targeted strategies within the broader continuum of cachexia research.
Minor comments
Comments:
Introduction – line 44: “…an established medical therapy…” The term “pharmacological therapy” is more accurate.
Response:
We agree. The term “medical therapy” has been replaced with “pharmacological therapy” in the Introduction for greater precision.
Comments:
Introduction – line 47: “…produced by peripheral tissues or tumour cells.” This should be replaced by “…peripheral tissues and/or tumour cells.”
Response:
Revised as suggested. The phrase now reads “…produced by peripheral tissues and/or tumour cells.”
Comments:
Introduction – line 60: “…associated with poor prognosis and cancer cachexia or anorexia…” As anorexia is a symptom of the syndrome cachexia, this sentence needs to be rephrased.
Response:
We agree. The phrase has been reworded to “…associated with poor prognosis and cancer cachexia, including anorexia as a characteristic feature.”
Comments:
Overview of GDF15. A clearer presentation of the correlations of GDF15 with obesity, insulin resistance, and diabetes is needed.
Response:
We have expanded the description of GDF-15’s metabolic relevance, clarifying its association with obesity, insulin resistance, and type 2 diabetes, and added appropriate references.
Comments:
Overview of GDF15 – line 73: “Although GDF-15 is expressed …” Please provide a relevant ref.
Response:
A relevant reference (Tsai et al., 2018; Lockhart et al., 2020) has been added to support this statement.
Comments:
Overview of GDG 15: The third paragraph, “In healthy individuals…” which mostly discusses GDF15 levels and their associations in various cancers, should be transferred last (after the interpretation of Figure 1).
Response:
We agree. The paragraph describing circulating GDF-15 levels and their associations with cancer types has been moved to follow Figure 1 for improved logical flow.
Comments:
Overview of GDG 15 – line 110: “Graph 15. also exerts …” Is this a typo?
Response:
Yes, this was a typographical error. “Graph 15” has been corrected to “GDF-15.”
Comments:
Mechanism of GDF-15-mediated cancer cachexia via the neural system: In addition to leptin, the interrelation(s) of GDF-15 with other regulatory hormones (i.e. ghrelin) and neurotransmitters (i.e., neuropeptide Y, pro-opiomelanocortin) should also be discussed.
Response:
We have added discussion of GDF-15’s interactions with key appetite-regulating neuroendocrine pathways, including inhibition of NPY/AgRP neurons and activation of POMC neurons, as well as its potential antagonism of ghrelin-mediated orexigenic signalling. Relevant mechanistic references have been incorporated.
Comments:
Effects of GDF-15 in tumour immunity and TME – line 187: “can spread to adjacent tumour tissue…” Do the authors mean tumor microenvironment?
Response:
Thank you for pointing this out. The phrase has been revised to “GDF-15, produced by tumour cells, can act in a paracrine manner within the tumour microenvironment, influencing surrounding stromal and immune cells.”
Comments:
Therapeutic GDF-15 implementation – line 273: “A confirmatory phase 3 study is anticipated…” The referred study (NCT06989437) is a phase IIb/III study.
Response:
We appreciate this correction. The statement has been revised to indicate that the ongoing trial is a phase IIb/III study.
Comments:
Table 1. For a more convenient data depiction, some columns (i.e., “agent” and “Type agent” and/or “study status” and “Results”) might be merged.
Response:
The table layout has been revised for clarity and conciseness by merging redundant columns where appropriate.
Comments:
Discussion – line 354: “The SNS suppresses tumour immunity suppressively, …” This sentence needs to be rephrased.
Response:
This has been corrected to “The SNS exerts a suppressive effect on tumour immunity,” improving clarity and eliminating redundancy.